# Ectopic Expression of *PsnNAC090* Enhances Salt and Osmotic Tolerance in Transgenic Tobacco

**DOI:** 10.3390/ijms24108985

**Published:** 2023-05-19

**Authors:** Yuting Wang, Wenjing Zang, Xin Li, Chaozheng Wang, Ruiqi Wang, Tingbo Jiang, Boru Zhou, Wenjing Yao

**Affiliations:** 1State Key Laboratory of Tree Genetics and Breeding, Northeast Forestry University, Harbin 150040, China; wyt1513026188@163.com (Y.W.); zangwenjing@163.com (W.Z.); lxngfu@163.com (X.L.); wangchaozheng2023@163.com (C.W.); yjswrq@outlook.com (R.W.); tbjiang@nefu.edu.cn (T.J.); 2Co-Innovation Center for Sustainable Forestry in Southern China, Nanjing Forestry University, 159 Longpan Road, Nanjing 210037, China; 3Bamboo Research Institute, Nanjing Forestry University, 159 Longpan Road, Nanjing 210037, China

**Keywords:** *Populus simonii × Populus nigra*, *PsnNAC090*, transgenic tobacco, salt stress, osmotic tolerance

## Abstract

The NAC transcription factor family is well known to play vital roles in plant development and stress responses. For this research, a salt-inducible NAC gene, *PsnNAC090* (Po-tri.016G076100.1), was successfully isolated from *Populus simonii × Populus nigra*. PsnNAC090 contains the same motifs at the N-terminal end of the highly conserved NAM structural domain. The promoter region of this gene is rich in phytohormone-related and stress response elements. Transient transformation of the gene in the epidermal cells of both tobacco and onion showed that the protein was targeted to the whole cell including the cell membrane, cytoplasm and nucleus. A yeast two-hybrid assay demonstrated that PsnNAC090 has transcriptional activation activity with the activation structural domain located at 167–256aa. A yeast one-hybrid experiment showed that PsnNAC090 protein can bind to ABA-responsive elements (ABREs). The spatial and temporal expression patterns of *PsnNAC090* under salt and osmotic stresses indicated that the gene was tissue-specific, with the highest expression level in the roots of *Populus simonii × Populus nigra*. We successfully obtained a total of six transgenic tobacco lines overexpressing *PsnNAC090*. The physiological indicators including peroxidase (POD) activity, superoxide dismutase (SOD) activity, chlorophyll content, proline content, malondialdehyde (MDA) content and hydrogen peroxide (H_2_O_2_) content were measured in three transgenic tobacco lines under NaCl and polyethylene glycol (PEG) 6000 stresses. The findings reveal that *PsnNAC090* improves salt and osmotic tolerance by enhancing reactive oxygen species (ROS) scavenging and reducing membrane lipid peroxide content in transgenic tobacco. All the results suggest that the *PsnNAC090* gene is a potential candidate gene playing an important role in stress response.

## 1. Introduction

Plants are often subjected to diverse environmental stresses such as low temperature, heavy metal, high salinity and osmotic stress in developmental processes [1,2]. In particular, high-salt and osmotic environments can cause physiological dehydration, nutrient imbalance and metabolic disorders in plants, seriously affecting growth and even causing death [3]. Plants adapt to abiotic stresses through many molecular and physiological mechanisms [4]. For example, plants can reduce the damage caused by reactive oxygen species (ROS) accumulation by improving the activity of antioxidant enzymes such as superoxide dismutase (SOD), catalase (CAT) and peroxidase (POD) [5,6].

Transcription factors (TFs) are important modulatory proteins that play key roles in a wide variety of biological processes [7,8]. At present, some important developmental processes have been introduced for deciphering the systemic TF networks in stress response in plants [9]. They are often involved in different metabolic and growth processes by binding to specific proteins and cis-elements in stress response pathways [9,10]. To date, there are a total of 58 TF families recorded in PlantTFDB (http://planttfdb.gao-lab.org/index.php, accessed on 13 March 2023) including MYB, ERF, WRKY, ZIP, HSF, NAC, etc. [11]. Among them, NAC is one of the largest plant-specific TF families that exerts an important role in plant growth and development and stress responses [12,13]. NAC family members have a consensus sequence known as the NAC domain (NAM, ATAF1, ATAF2 and CUC2) [14]. The NAC family has been recognized genome-wide in many plant species such as poplar (*Populus trichocarpa*), rice (*Oryza sativa*), maize (*Zea mays*) and Arabidopsis (*Arabidopsis thaliana*) with 150, 283, 152, 289 and 105 members, respectively [14,15]. NAC TFs are multifunctional proteins in diverse developmental events such as leaf senescence, shoot apical meristem development, flower development, hormone signaling and secondary wall formation. For example, overexpression of *RD26* in transgenic *Arabidopsis* exhibits a significantly reduced floral-tip dominance and has a positive role in ABA signaling [16,17]. Leaf senescence was delayed in transgenic plants overexpressing *VNI2*, whereas the opposite phenomenon was observed in VNI2-deficient mutants [18]. In addition, NACs have been proven to be associated with abiotic and biotic stress responses [19,20,21]. For instance, overexpression of *OsNAC10* promotes root growth and enhances drought tolerance, significantly increasing seed yield under drought conditions [21]. *OsNAC6* can be induced by abiotic stresses such as drought and high salt, and overexpression transgenic strains show greater tolerance to dehydration and high-salt stresses [22].

Poplar is widely used in China because of its high growth rate and strong adaptability [23]. However, its growth is also compromised by severe abiotic stresses, especially high salinity. In this research, we explored the biological function of a salt-induced NAC gene from *Populus simonii × Populus nigra*, *PsnNAC090*. We analyzed its subcellular localization, transcriptional activation region and upstream regulatory elements in the promoter sequence. We also demonstrated that the gene can specifically bind to ABRE elements. In addition, we obtained six transgenic tobacco lines overexpressing the gene through Agrobacterium-mediated transformation. Moreover, the transgenic tobacco lines showed higher resistance at the phenotypic and physiological levels under 200 mM NaCl and 20% PEG 6000 conditions. This study proves that *PsnNAC090* has a positive effect on plant salt-stress tolerance.

## 2. Results

### 2.1. Expression Pattern of NAC Family Genes under Salt Stress

To investigate the expression pattern of NAC family genes in the leaves of *Populus simonii × P. nigra* under salt stress, the per kilobase per million reads (FPKM) information of all 289 NAC TF genes was retrieved from the RNA-Seq dataset with NaCl treatment in 0 h, 12 h, 24 h and 36 h (Figure 1A). The differentially expressed genes (DEGs) were identified with an absolute value of log2FoldChange greater than 1 and a *p*-value less than 0.05 (Figure 1B and Appendix A). A total of 30 DEGs were screened to be shared at all time points of NaCl treatment. Among these DEGs, *PsnNAC090* was significantly salt-induced in the leaves and was selected for further functional validation.

### 2.2. Sequence Analysis of PsnNAC090

The sequence length of *PsnNAC090* is 765 bp, encoding 255 amino acids (aa), which has a highly conserved NAM domain in its N-terminus. According to NCBI blast, *PsnNAC090* has a 92.22% DNA sequence similarity with an NAC gene from *Populus trichocarpa* (Potri.016G076100.1, XP_002323414.1), and they share 3 highly conserved structural domains (Figure 2A). The protein is composed of 17.25% alpha helix, 14.12 extended chains, 3.14% beta bends and 65.49% random coils (Figure 2B). There were 8 proteins highly homologous to PsnNAC090 that were identified from *Populus trichocarpa* (XP_002323414.1), *Populus euphratica* (XP_011004699.1), *Ricinus communis* (EEF31871.1), *Salix brachista* (KAB5519700.1), *Jatropha curcas* (XP_037492117.1), *Hevea brasiliensis* (KAF2288696.1), *Gossypium gossypioides* (MBA0746825.1) and *Carya illinoinensis* (XP_042941525.1), with protein sequence similarities of 95.72%, 92.61%, 92.22%, 87.06%, 61.78, 63.04, 60.80% and 62.20%, respectively (Figure 2C).

In addition, we successfully obtained the upstream promoter sequence of *PsnNAC090* with a length of 1354 bp. This fragment is rich in typical regulatory elements (TATA-box, CAAT-box), jasmonic acid, abscisic acid, salicylic acid, gibberellin and phytohormone response elements (TGACG-motif, ABRE, as-1, P-box, MYB). It also contains a few other elements such as stress response elements (TC-rich repeats), light response elements (AAAC-motif, TCT, GT1-motif), inducer response elements (W-rich repeats) and phytohormone response elements (W-box) (Figure 2D and Appendix A).

### 2.3. Spatiotemporal Expression Pattern of PsnNAC090 in Populus

According to in silico prediction in PopGenIE (https://popgenie.org/gene?id=Ptri.016G076100, accessed on 3 February 2023), *PsnNAC090* was highly expressed in the roots and mature leaves, followed by the young leaves (Figure 3A). To validate the expression pattern of *PsnNAC090* in different tissues under abiotic stress, we subjected the one-month-old poplar seedlings to 200 mM NaCl and 20% PEG 6000 in 0 h, 3 h, 6 h, 12 h, 24 h and 48 h. RT-qPCR experiments revealed that the expression pattern of this gene was similar under both stresses, which was mainly highly expressed in the roots and leaves and relatively lowly expressed in the stems. Moreover, it peaked at 24 h in the leaves under salt stress and in the roots under 20% PEG 6000 stress (Figure 3B). An analysis of the RT-qPCR results suggests that the expression pattern of *PsnNAC090* under stress conditions is consistent with the in silico prediction for poplar.

### 2.4. Subcellular Localization of PsnNAC090 Protein

To explore the subcellular localization of PsnNAC090, we transiently transformed the recombinant vector 35S:PsnNAC090-GFP into *N. benthamiana* leaves with bacteriophage injection and onion epidermal cells with particle bombardment. Both results showed that the fluorescent signal of 35S:PsnNAC090-GFP appeared in the nucleus, cytoplasm and cell membrane, similar to the positive control, indicating that PsnNAC090 is a constitutive protein (Figure 4 and Appendix A).

### 2.5. Transcriptional Activation Activity of PsnNAC090

The full-length and segmented sequences of *PsnNAC090* were constructed into a pGBKT7 vector to obtain the recombinant vectors named pGBKT7-NAC090 (1–256aa), pGBKT7–NAC090a (1–136aa), pGBKT7-NAC090b (137–256aa), pGBKT7-NAC090c (197–256aa) and pGBKT7-NAC090d (167–256aa) (Figure 5A). Positive control (pGBKT7-53/pGADT7-T), negative control (pGBKT7) and recombinant plasmids were introduced into Y2H receptor cells. They were all found to grow normally on SD/-Trp medium, while only the positive control (pGBKT7), pGBKT7-NAC090, pGBKT7-NAC090b and pGBKT7-NAC090d grew normally and showed color reaction on SD/-Trp/-His/X-a-Gal medium (Figure 5B). PsnNAC090 was proven to have transcriptional activation activity, and the shortest function fragment was NAC090d (167–256aa).

### 2.6. Specific Binding of ABRE and PsnNAC090

The three repeat fragments of ABRE element (ACGTG) and *PsnNAC090* sequence were recombined into the pHIS2 reporter vector and the pGADT7-Rec2 effector vector, respectively (Figure 6A). Then, the negative control (pHIS2-p53/pGADT7-Rec2-NAC090), positive control (pHIS2-p53/pGADT7-Rec2-p53) and pHIS2-ABRE/pGADT7-Rec2-NAC090 were co-transferred into Y187 cells. The results show that they all grew normally in DDO medium, while only the positive control and pHIS2-ABRE/pGADT7-Rec2-NAC090 were able to grow normally on TDO/3-AT (50 mM) (Figure 6). This suggests that PsnNAC090 can specially bind to ABRE elements.

### 2.7. Germination Rate and Root Length of Transgenic Tobacco under Stress Conditions

A total of 100 seeds of transgenic tobacco were cultured on MS medium containing 20% PEG 6000 and 200 mM NaCl. There was no significant difference in the germination rate of wild-type (WT) and transgenic tobacco under control conditions. Under stress conditions, the germination rate of transgenic tobacco was significantly higher than that of WT tobacco. Under 20% PEG 6000 stress, the germination rate of transgenic tobacco was 1.16 times higher than that of WT tobacco. Under 200 mM NaCl stress, the germination rate of transgenic tobacco was 1.29 times higher than that of WT tobacco. This demonstrates that overexpression of *NAC090* improved the germination rate of transgenic tobacco under stress conditions (Figure 7B and Appendix A).

To observe the root growth of transgenic tobacco under stress conditions, the transgenic seedlings with similar growth state were transferred to MS containing 20% PEG 6000 and 200 mM NaCl (Figure 7A). The root length of the WT and transgenic seedlings displayed no significant difference under control conditions after 10 days of culture. However, under 20% PEG 6000 and 200 mM NaCl conditions, the root length of transgenic tobacco was 1.72 ± 0.03- and 1.49 ± 0.2-fold higher than that of WT tobacco, respectively (Figure 7B). We also investigated the expression pattern of *PsnNAC090* in the leaves of transgenic tobacco under stress conditions at 24 h. The results show that the relative expression level of *PsnNAC090* in transgenic lines was significantly higher than that in WT lines, and the *PsnNAC090* gene was induced by salt and osmotic stresses in the transgenic lines (Appendix A).

### 2.8. Histochemical Staining

To determine ROS accumulation in transgenic tobacco under stress conditions, we carried out NBT and DAB histochemical staining of transgenic tobacco leaves. The results suggest that there was no significant difference between WT and transgenic tobacco in staining degree under control conditions. However, the staining area and degree of WT tobacco were obviously higher than those of transgenic tobacco under stress conditions (Figure 8). This suggests that ROS accumulation in transgenic plants is obviously lower than that in WT plants.

### 2.9. Physiological Changes in Transgenic Tobacco under Stress Conditions

To further validate the function of *PsnNAC090*, we investigated the physiological parameters of 3 transgenic tobacco lines (T1, T3, T6) at 3 weeks old under 200 mM NaCl and 20% PEG 6000 conditions, including the changes in SOD activity, H_2_O_2_ content, chlorophyll content, POD activity, MDA content and proline content (Figure 9 and Appendix A).

SOD and POD, as important antioxidant enzymes, play an active role in helping plants scavenge excessive ROS. The results show that the SOD and POD activities of transgenic tobacco were significantly higher than those of WT tobacco under both stresses. Proline content in plants is an important physiological indicator of stress resistance. The accumulation of proline facilitates the regulation of intracellular osmotic pressure and high cellular water-retention capacity and reduces damage to cell membrane integrity. Chlorophyll is the pigment used for photosynthesis in plants, and a high chlorophyll content is more conducive to plant growth under adversity. In this study, both the proline content and chlorophyll content of transgenic tobacco were significantly higher than those of WT tobacco under salt and PEG stresses. In contrast, the MDA and H_2_O_2_ levels were lower in transgenic tobacco than in WT tobacco under both stresses. This suggests that ROS accumulation is higher in WT than in transgenic tobacco under stress conditions. Histochemical staining also validated this result. The results show that the overexpression of *PsnNAC090* improves the ability to scavenge ROS, increases the accumulation of proline and maintains the stability of cell membrane structure, thus enhancing the stress tolerance of transgenic plants.

## 3. Discussion

Plants suffer from various biotic and abiotic stresses during growth. Especially, salt and drought are important environmental factors affecting plant growth [24]. NAC genes are important TFs unique to plants and play an essential part in regulating plant growth and development [25,26]. For example, overexpression of *GmNAC20* and *AtNAC2* in transgenic *Arabidopsis* promotes lateral root formation [27,28,29]. Many NAC members have also been shown to function when plants are subjected to abiotic stresses; for example, *VuNAC1/2* is induced by various abiotic stimuli such as salt, osmotic stress and ABA, and overexpression of the gene in *Arabidopsis* promotes the growth of embryos, rosettes and inflorescences and improves tolerance to salt and osmotic stresses [30]. Overexpression of *SlNAC35* in tobacco was identified to promote root growth and development under drought and salt stresses [31]. In this study, we found that *PsnNAC090* was significantly upregulated in response to salt stress. We cloned the gene from *Populus simonii × P. nigra* and obtained transgenic tobacco lines overexpressing the gene using genetic transformation. Furthermore, we confirmed that overexpression of *PsnNAC090* significantly improved plant tolerance to high-salt and osmotic stresses at the phenotypic and physiological levels.

When plants face abiotic stress, a large amount of ROS accumulates in the body to poison them, so the ability to scavenge ROS is important for plants challenged with abiotic stress [31]. POD and SOD are important antioxidant enzymes beneficial for clearing excess ROS [32]. In this study, we measured the activities of POD and SOD in transgenic tobacco under high-salt and osmotic stresses. The results show that the transgenic tobacco plants have a stronger ability to scavenge ROS compared to the WT plants. In addition, we measured the H_2_O_2_ content in the transgenic tobacco under stresses, which indicated that there was less ROS accumulation in the transgenic tobacco than in the WT tobacco, which was also confirmed with NBT and DAB staining. These results indicate that *PsnNAC090* improves plant stress tolerance by reducing ROS accumulation.

MDA serves as an indicator of plasma membrane peroxidation, whose accumulation causes damage to the plasma membrane structure and affects normal cellular metabolism [33]. In addition, plants regulate intracellular osmotic pressure through the accumulation of proline to improve the water-retention capacity of cells [34]. In this study, the MDA content was lower and the proline content was higher in the transgenic tobacco than in the WT tobacco under high-salt and osmotic stresses. These findings indicate that *PsnNAC090* may enhance stress tolerance by maintaining the cell membrane and regulating intracellular osmotic pressure in transgenic plants.

ABA is an important signaling molecule in the abiotic stress pathway; it can be highly induced and accumulates when plants are subjected to high-salt and drought stresses [35,36]. The gene expression of many osmotic-related genes is regulated by NAC members through ABA-dependent pathways [37]. For example, *OsNAC2* directly downregulates the stress-related marker gene *OsSAPK1* through an ABA-dependent pathway under hypersalinity and osmotic stress [38,39]. *GhirNAC2* affects ABA biosynthesis and stomatal closure by regulating *GhNCED3a/3c* expression and improves drought tolerance in transgenic tobacco [40]. In ABA-dependent pathways, the cis-acting element ABA-responsive element (ABRE) has critical functions in gene expression [40]. For instance, *OsNAC5* regulates stress-related proteins by binding to ABREs under abiotic stresses, which enhances the salt tolerance of transgenic rice overexpression in the gene [41]. *SNAC-A* regulates leaf senescence signals and abiotic stress signals by binding to ABRE proteins [42]. In this study, we confirmed that PsnNAC090 can bind to an ABRE element. However, we are not sure which ABRE proteins can be regulated by PsnNAC090 or how PsnNAC090 can be involved in the ABA-signaling pathway, which questions need further validation in a future study.

## 4. Materials and Methods

### 4.1. Plant Materials

All the experimental materials were obtained from Northeast Forestry University, Harbin, China. *Populus simonii × P. nigra* seedlings were cultured in 1/2 MS (Murashige and Skoog) medium (pH 5.7) supplemented with 0.1 mg/mL indole 3-butytric acid (IBA). *Nicotiana benthamiana* were used for genetic transformation [43]. The tobacco seeds were sterilized with 20% bleach for 20 min and rinsed 5 times with sterile water. Then, they were sown uniformly on MS medium and grew to four true leaves. The tobacco seedlings were transferred to a clear culture flask containing MS medium and continued to grow for one month. An Agrobacterium-mediated leaf disc method was used for transgenic experiments.

### 4.2. Cloning PsnNAC090 and Its Promoter Sequence

RNA and DNA were extracted from *Populus simonii × P. nigra* leaves and used to clone *PsnNAC090* and its promoter sequence. The related kits were purchased from Takara, China. The sequence of *NAC090* (Potri.016G076100.1) was obtained from the Phytozome12 database (https://phytozome.jgi.doe.gov/pz/portal.html, accessed on 3 February 2023). The specific primer pairs including *PsnNAC090F1* and *PsnNAC090R1*, *PsnNAC090F2* and *PsnNAC090R2* were designed for gene and promoter cloning, respectively (Appendix A).

### 4.3. Analysis of PsnNAC090 and Its Promoter Sequence

Protein sequences with high homology to PsnNAC090 in eight different species were obtained from NCBI Database (http://www.ncbi.nlm.nih.gov/, accessed on 3 February 2023). Phylogenetic trees were constructed using the neighbor-joining method in MEGA 7.0. Prediction of cis-regulatory elements in the *PsnNAC090* promoter sequence was performed using PlantCRAE (http://bioinformatics.psb.ugent.be/webtools/plantcare/html/, accessed on 3 February 2023) and visualized using TBtools [44] and MEME Suite 5.3.0 (http://meme-suite.org/index.html, accessed on 5 February 2023) [45]. PsnNAC090 protein structure was predicted using SWISSMODEL (https://swissmodel.expasy.org/interactive/NHZXxy/models/, accessed on 5 February 2023).

### 4.4. Subcellular Localization of PsnNAC090

The CDS sequence without stop codon of PsnNAC090 was recombined into the pBI121-GFP vector driven with 35 s promoter to construct pBI121-PsnNAC090-GFP. The Agrobacterium tumefaciens containing the fusion constructs and pBI121-GFP vectors as positive controls were infused into the leaves of one-month-old *N. benthamiana* seedlings with bacteriophage injection. Meanwhile, the fusion vectors and control vectors were also introduced into onion epidermal cells with particle bombardment. The tobacco leaves and onion epidermal cells were then incubated in the dark for 36–48 h before observing their fluorescence signals through an LMS800 laser confocal microscope.

### 4.5. Transcriptional Activation of PsnNAC090

The full-length CDS sequence of *PsnNAC090* was divided into four segments, including NAC090a (0–136aa), NAC090b (137–256aa), NAC090c (197–256aa) and NAC090d (167–256aa). *PsnNAC090* and all the segments were inserted into the pGBKT7 vector to reconstitute pGBKT7-NAC090 (a-d) and pGBKT7-NAC090 fusion vectors. The above recombinant plasmids, pGBKT7 vector (negative control, and pGBKT7-53/pGADT7-T (positive control) were transferred into Y_2_H yeast receptor cells. Then, they were coated on SD/-Trp and SD/-Trp/-His/X-a-Gal medium and incubated at 30 °C for 3–5 days.

To verify whether PsnNAC090 can bind to ABRE (ACGTG) element, we performed yeast one-hybrid assay. We constructed the effector vector pGADT7-Rec2-NAC090 and the reporter vector pHIS2-ABRE and transferred negative control (pHIS2-p53/pGADT7-Rec2-NAC090), positive control (pHIS2-p53/pGADT7-Rec2-p53) and pHIS2- ABRE/pGADT7-Rec2-NAC090 into Y187 yeast cells. The successfully transformed yeast cells cultured in SDO liquid medium were adjusted to the same concentration and then diluted to 0×, 10×, 100× and 1000×. Moreover, they were respectively spotted on SDO and SD/-Trp/-Leu/-His/(TDO/3-AT (50 mM)) solid medium and incubated at 30 °C for 3–5 days. The primers used in this experiment are shown in Appendix A.

### 4.6. Spatiotemporal Expression Pattern of PsnNAC090

PopGenIE V3 database (https://popgenie.org/gene?id=Ptri.016G076100, accessed on 1 February 2023) was used to predict the expression pattern of *PsnNAC090* in different tissues in poplar. We treated one-month-old seedlings of *Populus simonii × P. nigra* with 200 mM NaCl and 20% PEG 6000 for 0, 3, 6, 9, 12, 24 and 48 h (3 biological replicates for each treatment). The leaf, stem and root tissues (three biological replicates for each sample) were collected after treatment. *Actin* gene was used as an internal reference gene for RT-qPCR, and the relative expression level of the gene was calculated using the 2^−ΔΔCt^ method [46]. The primers used for RT-qPCR are listed in Appendix A.

### 4.7. Acquisition and Identification of Transgenic Tobacco

An overexpression vector containing *PsnNAC090* was constructed using a homologous fusion method. The homologous arm containing a Spe1 digestion site was introduced at both ends of *PsnNAC090* and reconstituted with a pBI121 vector using infusion enzyme (bought from Takara). The recombinant vectors were transferred into GV3101 Agrobacterium tumefaciens. Positive strains were cultured with OD600 of 0.6 in LB liquid medium containing 50 mg/L rifampicin and 50 mg/L kanamycin. The leaves of three-week-old tobacco plants were cut to 1.0 × 1.0 cm and soaked in the bacterial solution for 10 min. Then, the leaf pieces were cultured in screening medium containing 50 mg/L kanamycin and 200 mg/L ceftriaxone sodium until resistant young shoots sprouted. Then, the resistant seedlings were obtained by culturing resistant shoots in MS medium containing 50 mg/L kanamycin and 200 mg/L ceftriaxone sodium. DNA was extracted from the leaves of resistant seedlings for molecular identification using specific primers of pBI121-GFP vectors (Appendix A). Three molecularly positive tobacco lines were randomly selected to collect their T_1_ seeds. Then, the T_1_ seeds were sterilized and screened in MS medium containing 100 mg/L kanamycin and 200 mg/L ceftriaxone sodium, and T_2_ and T_3_ seeds were obtained in the same way.

### 4.8. Stress Tolerance Analysis

T_3_ seeds were sterilized and spread on MS medium containing 200 mM NaCl and 20% PEG 6000 (each plate ≥ 80) for seed germination measurement. In addition, the transgenic tobacco seedlings with similar growth status were transplanted into rooting medium containing 200 mM NaCl and 20% PEG 6000 and continued to be cultured for 7 d for root length measurement (3 biological replicate trials).

### 4.9. Histochemical Staining and Physiological Measurement

Three one-month-old T_3_ transgenic tobacco lines were treated with 200 mM NaCl and 20% PEG 6000. After 4 h of treatment, the leaves were collected for tetrazolium blue chloride (NBT) and 3, 3′-diaminobenzidine (DAB) staining. After 5 days of treatment, the treated leaves were collected for physiological determination (3 biological replicates for each treatment). The kits for physiological indicators were purchased from Nanjing Jiancheng Institute of Biological Engineering, Nanjing, China.

## 5. Conclusions

In this study, we cloned *PsnNAC090* from *Populus simonii × P. nigra*, an NAC TF gene that was highly salt-induced in poplar leaves. PsnNAC090 was localized to the cytoplasm, nucleus and cell membrane. Its transcriptional activation activity region was located at 167–256aa. PsnNAC090 can bind to ABRE elements, which indicates that the gene participates in the ABA-signaling pathway. In addition, the promoter region of this gene is rich in phytohormone response elements, stress response elements and light response elements. A spatiotemporal expression analysis indicated that the gene was highly expressed in the roots and leaves under salt and osmotic stresses. The transgenic tobacco lines overexpressing the gene displayed morphological and physiological advantages under the stress conditions compared to the WT plants. The overall results indicate that *PsnNAC090* can enhance the salt and osmotic tolerance of transgenic tobacco.

## Figures and Tables

**Figure 1 ijms-24-08985-f001:**
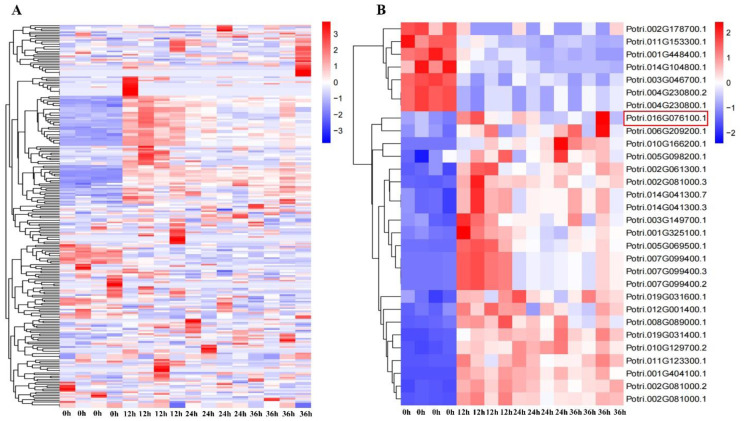
Heatmaps of NAC genes in the leaves of *P. simonii × P. nigra* under salt stress in 0 h, 12 h, 24 h and 36 h. (**A**) Expression patterns of 289 NACs under salt stress. (**B**) Expression patterns of 30 differentially expressed NAC members, with high and low expression levels represented in red and blue, respectively, and colored scales representing ploidy changes in transcript levels. Red box is the target gene.

**Figure 2 ijms-24-08985-f002:**
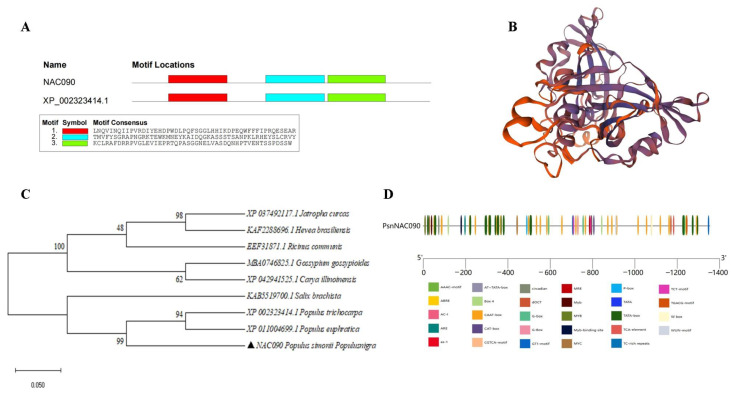
Sequence analysis of PsnNAC090. (**A**) Protein structure prediction of PsnNAC090. (**B**) Conserved structural domain composition of PsnNAC090. (**C**) Evolutionary analysis of PsnNAC090 and eight other homologous genes using neighbor-joining method in MEGA7. (**D**) Upstream promoter element analysis of PsnNAC090.

**Figure 3 ijms-24-08985-f003:**
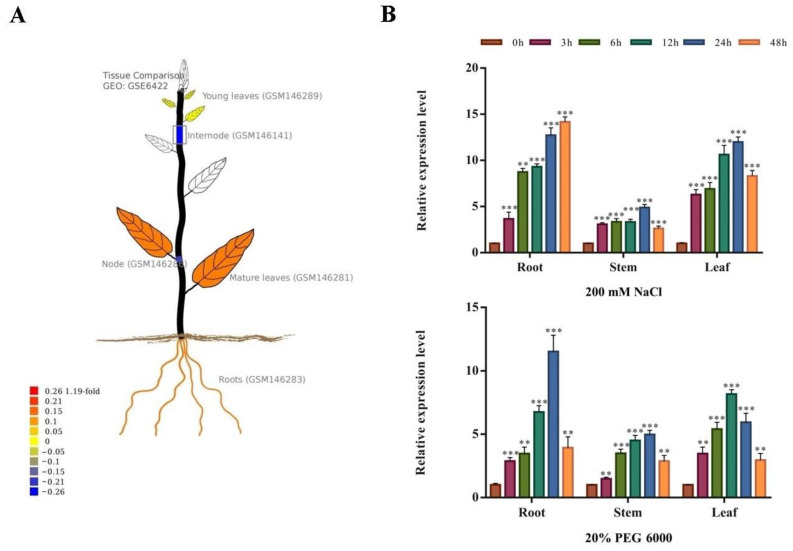
Spatiotemporal expression pattern of *PsnNAC090*. (**A**) Expression pattern of *PsnNAC090* in silico prediction in PopGenIE. (**B**) Relative expression levels of *PsnNAC090* in the roots, stems and leaves with treatments of 200 mM NaCl and 20% PEG 6000 in 0 h, 3 h, 6 h, 12 h, 24 h and 48 h (** *p* < 0.01, *** *p* < 0.001).

**Figure 4 ijms-24-08985-f004:**
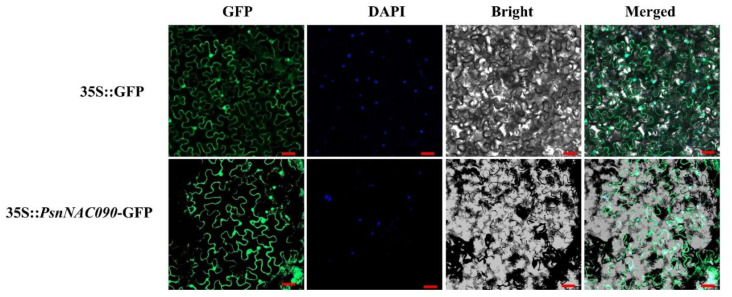
Subcellular localization of PsnNAC090 protein. (GFP) and (DAPI) are dark-field images, (Bright) is a bright-field image and (Merged) is a dark-field and bright-field superimposed image. Scale bar = 20 um.

**Figure 5 ijms-24-08985-f005:**
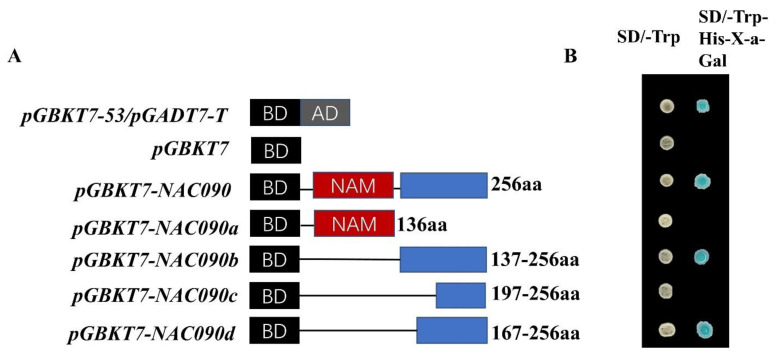
Transcriptional activation activity of PsnNAC090. (**A**) Schematic diagram of positive control (pGBKT7-53/pGADT7-T), negative control (pGBKT7) and segment sequences for pGBKT7-(1–305aa, 1–136aa, 137–256aa, 197–256aa, 167–256aa) vectors. The red squares represent conserved structural domains and the blue squares are non-conserved structural domain parts. (**B**) Yeast colony color reaction revealed pGBKT7-NAC090 (167–256) as the shortest transcriptional activation region.

**Figure 6 ijms-24-08985-f006:**
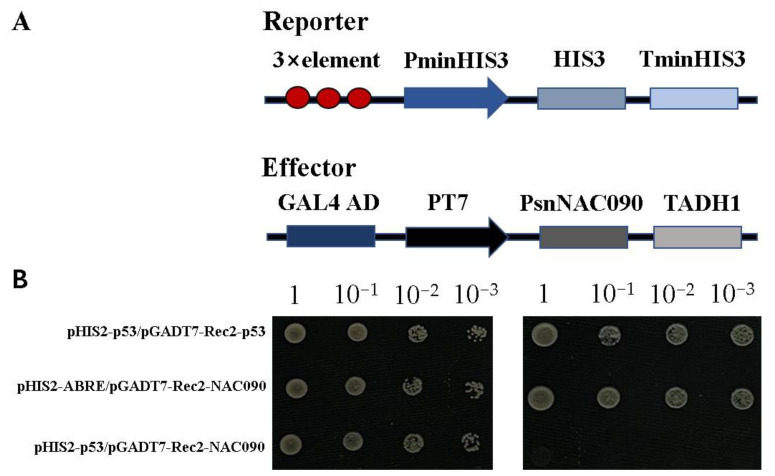
Specific binding of PsnNAC090 to ABRE. (**A**) Schematic diagram of reporting carrier and effector carrier. (**B**) Positive control (pHIS2-p53/pGADT7-Rec2-p53), negative control (pHIS2-p53/pGADT7-Rec2-PsnNAC090) and pHIS2-ABRE/pGADT7-Rec2-PsnNAC090 were co-transformed into Y2H yeast cells.

**Figure 7 ijms-24-08985-f007:**
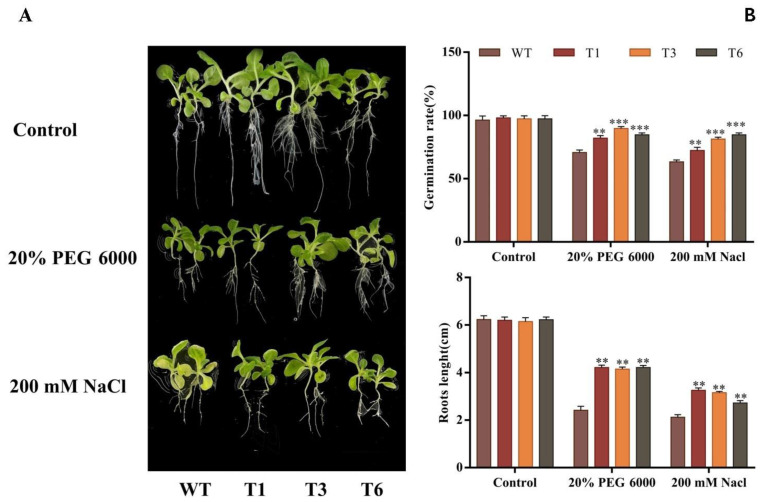
Growth state of transgenic tobacco under 20% PEG 6000 and 200 mM NaCl conditions. (**A**) Growth state of transgenic tobacco under stress conditions. (**B**) Germination rate and root length of transgenic tobacco under stress conditions. Transgenic lines and WT plants displayed significant differences indicated by asterisks in the error bars (*t* test, ** *p* < 0.01, *** *p* < 0.001).

**Figure 8 ijms-24-08985-f008:**
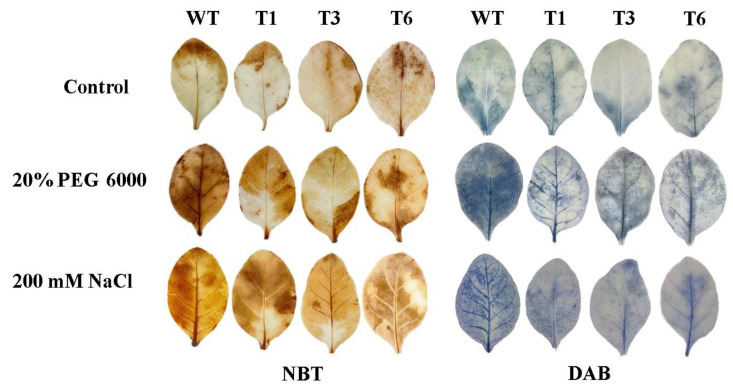
Nitroblue tetrazolium (NBT) and 3, 3′-diaminobenzidine (DAB) staining.

**Figure 9 ijms-24-08985-f009:**
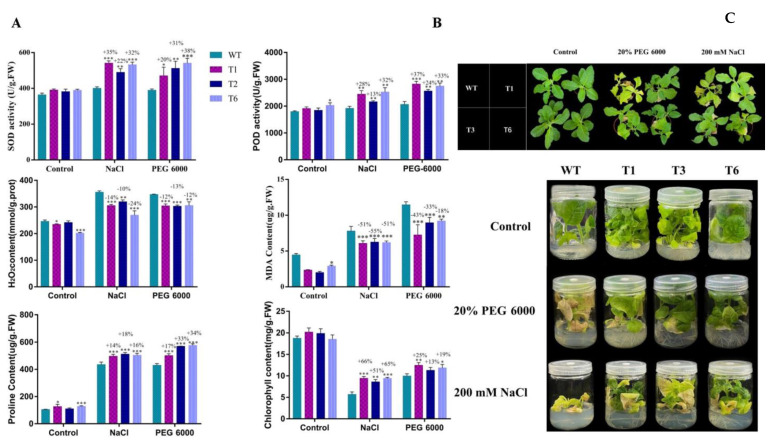
Physiological analysis of transgenic tobacco under 200 mM NaCl and 20% PEG 6000 stresses. (**A**) Physiological indicators including SOD activity, POD activity, H_2_O_2_ content, MDA content, proline content and chlorophyll content. (**B**) Phenotype comparison of transgenic tobacco and WT tobacco at three weeks old in soil pots under stress conditions. (**C**) Phenotype comparison of transgenic tobacco and WT tobacco in MS medium containing 200 mM NaCl and 20% PEG 6000. Three biological replicates were prepared for each group; error bars indicate mean ± SD (*t*-test, * *p* < 0.05, ** *p* < 0.01, *** *p* < 0.001).

## Data Availability

Not applicable.

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
