# Peer review of "Ectopic Expression of PsnNAC090 Enhances Salt and Osmotic Tolerance in Transgenic Tobacco"

_ijms, 2023, doi:10.3390/ijms24108985_

Round 1

Reviewer 1 Report

At present ms, the author isolated a salt-induced NAC gene (PsnNAC090) from hybrid poplar and ectopic expressed in transgenic tobacco plant to clarify its possible function on abiotic stress tolerance. Basically, the experimental design and result are logical and reasonable, however, many inconsistent description and error in this ms. They have to revise very detail and carefulness. As follow:

1. The activation domain of PsnNAC090  promoter is confused, in Abstract, line 9, 169-256 aa., in Figure 5 is 167-256 aa., in Conclusion, is 197-256 aa.?

2. The PEG-6000, somewhere described PEG-2000, ex. Figure 7, Figure 9, Material and Methods, and whole ms.

3. ABA response element (ABRE), the full sentence should be presented at the first time mentioned, however, it showed only in Page 11, most seriously, the abbrev. sometimes ABER (ex. Page 2), sometimes AERE (ex. Page 6)....Totally messy! 

4. Sodium Chloride, NaCl, many typo in this ms to NACL or Nacl,....

5.The authors should provide complete plasmid construction information for tobacco transformation. Ex, promoter design? 

6. in Page 8, Section 2.8.  the description about NBT and DAB staining should be Figure 8, not Figure 9.

7. All the "control" in the Figure or table should be "Control".

8.The authors should explain why this transcription factor, PsnNAC090,  expressed in whole cell by subcellular location analysis!

9. The authors should provide the data or description about the endogenous tobacco NAC090 related gene expression pattern under control and stresses condition.

10. According the Figure 9, the SOD, and POD activities from transgenic tobacco lines in control condition are higher than wild type, and the enhanced activities just similar to wild type under salt and PEG treatment, that mean the enzyme activities should enhance by stresses treatment not caused by overexpressed PsnNAC090 gene.

The English writing is fine.

Reviewer 2 Report

The research paper has a certain degree of innovation, and the research ideas are relatively clear and reasonable. However, there are still many areas that need to be modified in the paper, and further experiments need to be supplemented to make the content of the article more rigorous. 

1、The chart in the paper is not clear enough, it is recommended to improve the resolution. 

2、The subcellular localization results are not accurate enough. It is recommended to supplement experiments, especially by using different colors of fluorescence and co expression of 35s GFP, to determine the subcellular location of this protein in detail.

3、It is suggested to use transgenic plants, transcriptome sequencing and other technologies to study the downstream regulatory gene of this protein under drought and salt treatment, which can make the article more complete.

Chart production and the English proficiency of the paper require in-depth revisions. 

Round 2

Reviewer 1 Report

The authors have carefully and completely revised this manuscript. Only very minor revision is necessary, ex.

1. 200mM NaCl should be 200 mM NaCl,  line 540, H2O2 should be H2O2....

2. Page14, line 415, Agrobacterium tumefaciens should be italic.
